# Management of Osteoarthritis and Joint Support Using Feed Supplements: A Scoping Review of Undenatured Type II Collagen and *Boswellia serrata*

**DOI:** 10.3390/ani13050870

**Published:** 2023-02-27

**Authors:** Ana Zapata, Rocio Fernández-Parra

**Affiliations:** 1Hospital Veterinario de Referencia UCV, Departamento de Medicina y Cirugía Animal, Facultad de Veterinaria y Ciencias Experimentales, Universidad Católica de Valencia San Vicente Mártir, 46018 Valencia, Spain; 2Escuela de Doctorado, Programa de Doctorado en Ciencias de la Vida y del Medio Natural, Universidad Católica de Valencia San Vicente Mártir, 46002 Valencia, Spain; 3Departamento Medicina y Cirugía Animal, Facultad de Veterinaria y Ciencias Experimentales, Universidad Católica de Valencia San Vicente Mártir, 46002 Valencia, Spain

**Keywords:** osteoarthritis, Undenatured type II collagen, UC-II^®^, *Boswellia serrata*, feed supplements, joint inflammation, joint degeneration

## Abstract

**Simple Summary:**

Undenatured type II collagen and *Boswellia serrata* are feed supplements that have been used for years in the multimodal management of osteoarthritis (OA), with the aim of maintaining articular cartilage and reducing the inflammatory state. The objective of this review is to analyse how the administration of undenatured type II collagen and *Boswellia serrata* or both combined in the same product influences the management of OA and helps support joint health and mobility. Twenty-six articles were selected, which were carefully analysed. It was observed, based on different methods, that the use of these two feed supplements is a valid option for the long-term multimodal management of OA, improves the general condition, appetite and mobility, and reduces the degree of lameness. Its use before intense exercise is associated with greater activity and less response from inflammatory biomarkers.

**Abstract:**

In the multimodal management of osteoarthritis (OA) in recent decades, the use of feed supplements to maintain joint cartilage has been advocated. The aim of this scoping review is to present the results found in the veterinary literature on the use of undenatured type II collagen and *Boswellia serrata* in dogs, specifically its use in dogs with clinical signs of OA, healthy dogs after intense exercise or dogs with diseases that predispose the individual to OA. For this purpose, a literature review was carried out using the electronic databases PubMed, Web of Science and Google Scholar, from which a total of 26 records were included in this review: fourteen evaluating undenatured type II collagen, ten evaluating *Boswellia serrata* and two evaluating the combination of undenatured type II collagen and *Boswellia serrata*. The review of the records showed that undenatured type II collagen decreases the clinical signs associated with OA, improving the general clinical state with a reduction in the degree of lameness and increase in physical activity or mobility. Evaluating the response to supplementation with *Boswellia serrata* alone is complicated due to the limited publication of studies and variations in the purity and compositions of the products used, but in general terms, its combination with other feed supplements produces benefits by relieving pain and reducing the clinical signs of OA in dogs. The combination of both in the same product provides results similar to those obtained in undenatured type II collagen studies. In conclusion, undenatured type II collagen and *Boswellia serrata* are considered a valid option for the multimodal approach to the management of OA and for improving activity during intense exercise, but more studies are needed to conclude whether or not it prevents OA in dogs.

## 1. Introduction

Osteoarthritis (OA) is a slowly progressive and dynamic degenerative disease that affects mobile joints. The pathophysiology of OA has been considered a process of prolonged and repeated wear and tear of the articular cartilage with alterations in subchondral bone metabolism (sclerosis) and new bone formation at the joint margins (osteophytes) [1,2]. Today, it is considered a more complex process, with metabolic and inflammatory factors. It is known that the accumulation of articular cartilage matrix degradation products stimulates the release of inflammatory mediators, such as cytokines (IL-1β, IL-6 and TNF-α) and prostaglandin E2 (PGE-2), from the synovium and subchondral bones. These factors act on cartilage by inhibiting proteoglycan synthesis and stimulating its destruction, creating a feedback loop [3]. On the other hand, the endocrine system regulates chondrocyte activity, meaning that many growth factors are responsible for regulating chondrogenesis, such as insulin growth factor (IGF-I) and transforming growth factor β (TGF-β), which play key roles in chondrocyte maturation and survival and matrix formation, limiting catabolism by antagonising pro-inflammatory substances [4]. The means by which such biochemical and mechanical processes contribute to the progressive failure that characterises OA are closely related to the interaction between joint damage, the immune response to perceived damage and subsequent chronic inflammation [3]. The presence of OA in dogs is mostly secondary to a developmental or acquired pathology [2,5]. The prevalence of OA differs in the veterinary literature, ranging from 2.5% to 20% for dogs older than one year [6,7] and increasing to more than 80% for dogs older than eight years [1,8]. Even though OA has been found to have a highly negative impact on patient well-being when compared to other common diseases [9,10], a definitive curative treatment has not been identified. For this reason, the strategies used for managing OA are broad, and multimodal management of the disease is generally carried out. Once OA is established, owners must be aware that it is a lifelong disease. It cannot be cured, and the clinician’s main objective is to relieve pain, protect the joint from the disease’s progression, and establish proper nutritional support, which will improve mobility and allow normal activity to be maintained or regained [11]. For this reason, the management of OA combines pharmacological treatments with non-pharmacological treatments.

Non-steroidal anti-inflammatory drugs (NSAIDs) are the usual treatment par excellence for OA [12] and can be used in the long term. However, their use is associated with gastrointestinal effects and hepatic and renal injury [13,14]. Other drugs that can be combined in instances of acute pain and are synergistic with NSAIDs are paracetamol (acetaminophen) [15] and tramadol, although the bioavailability and half-life of the latter varies between individuals and is therefore not recommended for solo use [16,17]. Tapentadol appears to have a superior analgesic profile, but further studies are required to assess its efficacy in patients with OA [18]. Chronic OA pain has a maladaptive component caused by central sensitisation; hence, drugs such as gabapentin [19], amantadine [20] and tricyclic antidepressants (amitriptyline and nortriptyline) [21] may be chosen for neuropathic pain control. In recent years, cannabinoid therapy has been considered an option, given that some studies have associated it with an improvement in quality of life [22,23,24]. Recently, species-specific anti-nerve growth factor monoclonal antibodies have been used as a novel therapy for chronic pain associated with OA in dogs, with positive results obtained for at least three months, although more studies are needed to evaluate the duration and analgesic effects [25].

In addition to pain control, feed supplement products are recommended with the aim of maintaining joint cartilage. These feed supplements may reduce the inflammatory state and oxidative stress on the cellular level, acting through different mechanisms of action [26,27,28]. The most common oral administration products used for OA in veterinary care are undenatured type II collagen [29], glucosamine (GLU), chondroitin (CHO) [30], green-lipped mussels [31] and omega-3 fatty acids [32,33]. Other feed supplements based on medicinal plants have been described, such as *Boswellia serrata* [34], *Ribes nigrum*, Shiitake (*Lentinus edodes*), *Harpagophytum procumbens* and Curcuminoid extracts [26,28]. An innovative therapeutic approach that combines undenatured type II collagen with *Boswellia serrata* has provided positive results in patients with degenerative joint diseases in the case of both humans and animals [35,36,37]. 

Undenatured type II collagen is a non-hydrolysed collagen which is extracted from the chicken sternum through a very limited mechanism that allows its native triple helix conformation to be preserved, presenting with a molecular weight of 300 kDa [38]. It must be differentiated from hydrolysed (denatured) collagen, which is broken down into different peptide components by means of enzymes, heat or pH [39] and has a molecular weight of 2 to 9 kDa [38]. In humans, preclinical studies and in vivo clinical trials arrived at conclusions on the beneficial effects of collagen derivatives for OA and cartilage repair used as a nutritional supplement. However, in a large portion of the available in vitro studies, hydrolysed collagen from different sources and of different molecular weights was ineffective for, or even detrimental to, OA cartilage [40]. Undenatured type II collagen is believed to act through an oral tolerance mechanism, thus initiating anti-inflammatory and protective cartilage pathways that prevent the immune system from damaging its own articular cartilage [41]. Oral tolerance is an immune process that the body uses to distinguish harmless material in the gut, such as dietary proteins and the commensal organisms that comprise the microbiome, from potentially harmful foreign invaders [42,43]. This detection is performed by the gut-associated lymphatic tissue (GALT), which is composed of mesenteric lymph nodes and patches of lymphoid tissue that surround the small intestine, called Peyer’s patches, which absorb and select compounds from the intestinal lumen and, depending on the compound, activate or deactivate the body’s immune response through the gastrointestinal tract [44]. Laboratory studies on rats have indicated that Undenatured Type II Collagen orally ingested at low doses is transported through intestinal epithelial cells to underlying immune cells in Peyer’s patches, where naive T cells transform into regulatory T (Treg) cells that are specifically activated by type II collagen. The activated Treg cells then migrate from the GALT through the lymphatic system and enter the bloodstream [45]. After recognising their target (type II collagen) in the articular cartilage, Treg cells secrete anti-inflammatory cytokines such as TGF-beta, interleukin (IL)-4 and IL-10. This action suppresses the actions of cells involved in the normal degradation of collagen and other extracellular matrix proteins, resulting in a reduction in joint inflammation and associated discomfort [46]. The mechanism by which oral tolerance is activated takes at least two to three weeks and may take up to three months in order to be fully effective [47].

*Boswellia serrata* is a tree that is widely distributed in India, and its oily gum resin has traditionally been used for centuries in humans as a remedy to treat inflammatory diseases [48]. The resinous part of *Boswellia serrata* possesses monoterpenes, diterpenes, triterpenes, tetracyclic triterpenic acids and four major pentacyclic triterpenic acids, i.e., β-boswellic acid, acetyl-β-boswellic acid (AβBA), 11-keto-β-boswellic acid and acetyl-11-keto-β-boswellic acid (AKBA) [49]. Boswellic acids with the characteristic pentacyclic triterpene ring can exhibit actions related to the inflammatory cascade, with AKBA being more active, selectively inhibiting a branch of the arachidonic acid cascade (5 LOX) related to the production of leukotrienes without affecting other activities of LOX and COX. The inhibition of 5 LOX and leukotriene synthesis reduces the levels of pro-inflammatory cytokines, leading to decreased cartilage destruction and inflammatory cell chemotaxis and producing a balance in favour of cartilage regeneration [50,51].

A review of UC-II^®^ for the management of OA in companion animals has been published in the veterinary literature. It refers to an undenatured type II collagen patented under a registered trademark [29]. However, it is not clear what type of information is available in the literature on the use of undenatured type II collagen or whether the results are similar to those of UC-II^®^. In addition, the authors are currently unaware of any reviews in the literature on the use of *Boswellia Serrata* alone or in combination with undenatured type II collagen for the management of OA or for keep joint healthy in dogs. For these reasons, a scoping review was carried out regarding the use of undenatured type II collagen and *Boswellia serrata* as feed supplements in the management of OA and for helps support joint health in dogs. The following research question was formulated: What is known from the literature on the use of products containing Undenatured Type II Collagen and/or *Boswellia serrata* for keep joint healthy or amelioration of clinical signs of OA in dogs?

## 2. Materials and Methods

A keyword search of the literature was performed to identify all the available existing publications related to the use of Undenatured Type II Collagen and *Boswellia serrata* for managing clinical signs of OA and helping support joint health in dogs in electronic databases commonly used in veterinary medicine: PubMed/MEDLINE (https://pubmed.ncbi.nlm.nih.gov (accessed on 29 January 2023)) and Web of Science (Web of Science Core Collection, MEDLINE, Current Contents Connect and Derwent Innovations Index) (https://apps.webofknowledge.com (accessed on 29 January 2023)). The research strategy was performed with keywords linked through Boolean terms. Similar terms for “Undenatured type II collagen and *Boswellia serrata*” were included by combining them with synonyms for “osteoarthritis” in dogs, as follows: (“undenatured type II collagen” OR “type II chicken collagen” OR CCII OR UCII OR “undenatured collagen type II” OR “*Boswellia serrata*” OR “*Boswellia serrata* gum resin extracts” OR “*Boswellia serrata* extract” OR “boswellia extract” OR “boswellic acid” OR “boswellia resin”) AND (Osteoarthritis OR osteoarthrosis OR osteoarthroses OR osteoarthrosic OR arthrosis OR arthritis OR arthritic OR “degenerative arthritis” OR “degenerative joint disease” OR “degenerative joint disease djd” OR “cartilage degeneration” OR “inflammatory joint” OR “joint diseases”) AND (dog OR canine). Whenever possible, both the singular and the plural forms were used. A language filter was applied, including only articles in English and Spanish. PubMed was searched by title/abstract, and the Web of Science search was performed by topic. In addition, a search was carried out in Google Scholar for the first 100 records that were obtained. Due to the maximum number of keywords allowed by the search engine, the following search strategy was used: (“undenatured type II collagen” OR UCII OR “undenatured collagen type II” OR “*Boswellia serrata*” OR “boswellic acid”) (osteoarthritis OR osteoarthrosis OR arthrosis OR “degenerative joint disease” OR “cartilage degeneration”) (dog OR canine). 

The inclusion criteria for each publication were as follows: (1) the use of undenatured type II collagen, *Boswellia serrata* or a commercial product containing these active ingredients in the study; (2) evaluation of their effects in managing clinical signs of OA or for helps support joint health using the following methods: owner and veterinarian questionnaires, chronic pain assessment scales, general clinical assessment scales (appetite, mobility and lameness) and quantitative methods using force plate gait and inflammatory biomarkers in the blood or synovial fluid; (3) oral application; and (4) studies conducted on dogs. The exclusion criteria were as follows: (1) reviews, book chapters, expert opinions of clinicians, authorities and/or reports of expert committees; (2) studies evaluating only alterations on the cellular, microbiological or histological level; and (3) studies measuring only plasma concentrations of undenatured type II collagen or *Boswellia serrata*. The titles and abstracts of all the records retrieved from the search were manually reviewed, and for records that used the trade name of the product or did not specify the feed supplement used for the study, we read the entire record to determine whether it met the inclusion criteria. A double-screened title and abstract analysis was performed using the results obtained from the search in PubMed and Web of Science, and the level of agreement between authors was 95.1%. 

The principal investigator searched and reviewed all the records. The data extracted from the studies included the date of publication, type of study design, whether or not the study was blind (owner or veterinarian), double-blind (veterinarian and owner) or not blind, the feed supplements used, adverse effects and results obtained through owner or veterinarian questionnaires, chronic pain assessment scales, general clinical assessment scales (appetite, mobility and lameness), quantitative methods (using force plate gait, an accelerometer and a GPS collar), radiographic or ultrasonographic examination and inflammatory biomarkers in the blood or synovial fluid. The records were classified according to the design of the scientific study, as described in Table 1. The scoping review is organised into three different blocks in the Results and Discussion sections. The first part compiles information on the use of undenatured type II collagen, the second describes the use of *Boswellia serrata*, and finally, the review focuses on the use of the combination of undenatured type II collagen and *Boswellia serrata*.

## 3. Results

The search strategy resulted in a total of 49 records published up to 29 January 2023. The PubMed search showed 18 articles, and Web of Science yielded 31 articles. After eliminating duplicate articles (*n* = 16) and two articles that were leaflets or patents, the titles and abstracts of the rest of the records were reviewed, excluding those that did not meet the inclusion and exclusion criteria from the review (*n* = 8). In the search carried out with Google Scholar, four new records were found (three articles and a congress proceeding) [26,36,52,53]. An article cited in the bibliographical references of the records was added to the review [54]. In total, 28 bibliographic references were obtained for the complete review of the full text. There were three records from conference proceedings; hence, only the abstract was taken into account for the results and discussion. One congress proceeding [53] was subsequently excluded after reading, since it provided the same information as one of the articles already included in the review, and one study was conducted in humans; hence, it was excluded. A flow diagram of the literature search and study selection process is presented in Figure 1.

From the data collected on the feed supplements used in the studies, it was recorded that fourteen of them evaluated undenatured type II collagen alone, one of them evaluated *Boswellia serrata* alone, nine of them evaluated *Boswellia serrata* together with other feed supplements, and two evaluated the combination of undenatured type II collagen and *Boswellia serrata* (Appendix A). The data collected on the year of publication, the type of study design and whether the study was blinded or not are collected in Table 2, Table 3 and Table 4 for undenatured type II collagen, *Boswellia serrata* and undenatured type II collagen combined with *Boswellia serrata*, respectively. In total, 73% of the articles were prospective randomised placebo-controlled clinical trials, 12% were prospective randomised controlled clinical trials, and 15% were prospective clinical trials. Of the fourteen studies that evaluated the results after supplementation with undenatured type II collagen, 79% of them were prospective randomised placebo-controlled clinical trials, 14% were prospective randomised controlled clinical trials, and 7% were prospective clinical trials. Of the ten studies evaluating the use of feed supplements with *Boswellia serrata*, 70% were prospective randomised placebo-controlled clinical trials, 20% were prospective randomised controlled clinical trials, and 10% were prospective clinical trials. Of the two studies evaluating the combination of undenatured type II collagen and *Boswellia serrata*, only one study was a prospective randomised placebo-controlled trial [37], while the other one was a clinical trial and did not have a control group [36].

Of the total articles reviewed, 21 of the studies used undenatured type II collagen and *Boswellia serrata* for the management of OA, and 5 of them used this feed supplements for helps support joint health after intensive exercise or in patients with diseases that predispose to OA, such as hip and elbow dysplasia or medial patellar luxation [52,61,62,63,66].

### 3.1. Undenatured Type II Collagen

The first study to evaluate the effects of UC-II^®^ in managing OA in dogs was conducted in 2005, comparing two groups of dogs supplemented with doses of 1 mg/day and 10 mg/day of active ingredient UC-II^®^ with a placebo group. The results showed a decrease in overall patient pain, pain after manipulation and the degree of lameness after exercise in the two supplemented groups after 90 days [55]. Subsequent studies conducted by the same research group evaluated the effects of active ingredient UC-II^®^ at a dose of 10 mg/day, either alone or in combination with other feed supplements, over longer periods (120–150 days) on dogs with OA [30,56,57,58]. They used observational numerical scales from 0 to 10 (0 no pain, 5 moderate and 10 severe and constant pain) to assess overall pain and scales from 0 to 4 (0 no pain, 1 mild, 2 moderate, 3 severe, 4 severe and constant) to assess the pain response after the manipulation of the limbs in flexion and extension, as well as lameness after exercise [30,56,58]. The results showed that daily use of UC-II^®^ reduced the level of generalised pain by 33% within the first month, with a 44% reduction in exercise-associated lameness and a 66% reduction in pain on manipulation within the first two months [30]. They observed that the maximum benefit occurred 4–5 months after the start of supplementation, with a 62–81% reduction in generalised pain, 83–91% reduction in pain on limb manipulation and 78–90% reduction in exercise-associated lameness [30,56,57]. The results obtained with NEXT-II were similar [59], although the reduction in generalized pain was not as evident as that observed for UC-II^®^. The combination of GLU and CHO with UC-II^®^ did not provide a greater reduction in generalised pain, pain upon limb manipulation or exercise-associated lameness compared to the reduction obtained with the single administration of UC-II^®^ [30]. The combination of UC-II^®^ with HCA and CM seemed to show improved benefits through the reduction in overall pain, pain upon limb manipulation and exercise-associated lameness [56]. The management of OA with UC-II^®^ was analysed in two studies comparing the use of Robenacoxib and Cimicoxib, and there were no significant differences [60,65]. The pain assessment questionnaires completed by the owners and veterinarians showed favourable results after the use UC-II^®^ [60,65,66,67]. The results obtained using quantitative methods for the gait analysis showed an improvement in step force [57,58,66]. In healthy dogs subjected to intense exercise, supplementation with UC-II^®^ led to increased speed of movement and activity per kilometre [63]. On the blood level, researchers observed increased lymphocytes; decreased neutrophils, eosinophils and basophils; and decreased biomarker IL-6 and cartilage oligomeric matrix protein [62,64,66]. Long-term supplementation with UC-II^®^ showed no adverse effects or significant changes in hepatic or renal function [30,55,56,57,58,59], except for blood urea nitrogen in one study [61]. The results are summarized in Table 5. 

### 3.2. Boswellia serrata

The only study that explicitly evaluated the use of *Boswellia serrata* was performed on dogs with chronic inflammatory and degenerative joint and spine disease. In this study, the authors administered a daily dose of *Boswellia serrata* (the extract contained ≥50% triterpenic acids) at 400 mg per 10 kg and observed a beneficial response in 71% of patients. They showed a reduction in clinical signs after two weeks, such as intermittent lameness, lameness after a prolonged rest, local pain and stiffness when walking [34]. The use of *Boswellia serrata* extracts has been combined with other natural herbal products (*Harpagophytum procumbens*, *Ribes nigrum*, *Ananas comosus* and *Curcuma longa*) and feed supplements (omega-3 fatty acid, GLU, methylsulfonylmethane, CHO and L-glutamine) showing improvements in all clinical sings (lameness, pain on manipulation, range of motion and joint swelling) [26,68]. The results obtained using quantitative methods for gait analysis show an improvement in step force [69,70]. Conflicting results between studies have been observed using assessment questionnaires completed by the owners and veterinarians [36,69,71,72,73]. The use of feed supplements with *Boswellia serrata* has been described in growing puppies, and the results seem to show that although it does not prevent OA induced by these pathologies, a lesser degree of OA is evident at 12 months of age [52]. It has been observed that the use of *Boswellia serrata* together with other feed supplements for patients with OA can result in a decrease in alkaline phosphatase, cholesterol and triglycerides [26], but no significant change was observed for haematological samples [26,34,71,72]. In addition, changes in OA biomarkers have been observed, including increased IL-10 and Glutathione and decreased IL-6, IL-2 and C-reactive protein [26,69,71,72]. Gastrointestinal disturbances were reported as adverse effects in two studies [34,73]. The results are summarized in Table 6.

### 3.3. Undenatured Type II Collagen and Boswellia serrata

One product is described in the literature that combines undenatured type II collagen and *Boswellia serrata* together: Confis Ultra (Candioli Pharma S.r.l., Beinasco, Italy). In the studies published by Martello and colleagues, the authors used one tablet (2 g) of Confis Ultra (Candioli Pharma Srl.) for every 10 kg of weight [36,37]. The tablets contain 4 mg of undenatured type II collagen (Producer, city, state abbr. if Canada or USA, country), 31.5 mg of *Boswellia serrata*, *Camellia sinensis*, green tea extract, copper complexes of chlorophylls (Producer, city, state abbr. if Canada or USA, country), CHO, GLU and Hyaluronic acid.

The combined supplementation, after 60 days, showed an improvement in the clinical signs of the patients associated with pain, with lameness being reduced in 69% of patients. Its use seemed to be more effective for more severe degrees of OA, as the owners reaffirmed that after 60 days of supplementation, their pets’ mood, lameness and mobility improved, but no such significant difference was observed at 30 days. No adverse effects or analytical alterations were observed [36,37]. The results are summarized in the Table 7.

**Table 5 animals-13-00870-t005:** Literature overview of evidence regarding undenatured type II collagen for the management of osteoarthritis (OA) and joint support in dogs.

Reference	Nº Animals, Groups, and Duration	Objectives	Main Results y Conclusion
Deparle et al. 2005 [55]	Fifteen client-owned dogs with OA - Placebo (Group I) - UC-II^®^ (Group II) - UC-II^®^ (Group III) 90 days	Evaluate overall pain, pain during limb manipulation and exercise-associated lameness	Groups II and III showed a decline overall pain, pain during limb manipulation and lameness after physical exercise (*p* < 0.05)
D’Altilio et al. 2006 [30]	Twenty client-owned dogs with OA - Placebo (Group I) - UC-II^®^ (Group II) - GLU + CHO (Group III) - UC-II^®^ 10 + GLU + CHO (Group IV) 120 days	Evaluate overall pain, pain during limb manipulation and exercise-associated lameness	Group II showed a reduction in overall pain (62%), pain upon limb manipulation (91%) and exercise-associated lameness (78%) (*p* < 0.05) Group IV showed a maximum reduction in overall pain (57%), pain upon limb manipulation (53%) and exercise-associated lameness (53%) (*p* < 0.05) after 120 days
Peal et al. 2007 [56]	Twenty-five client-owned dogs with OA - Placebo (Group I) - UC-II^®^ (Group II) - HCA (Group III) - HCA + CM (Group IV) - UC-II^®^ + HCA + CM (Group V) 120 days	Evaluate overall pain, pain during limb manipulation and exercise-associated lameness	Groups II and V showed reductions in overall pain (62–70%), pain upon limb manipulation (67–91%) and exercise-associated lameness (69–78%) (*p* < 0.05 Group IV showed a significant reduction in pain (*p* < 0.05)
Bagchi et al. 2009 [57]	Dogs with OA (no information about sample size) - Placebo (Group I) - UC-II^®^ (Group II) 120 days	Evaluate overall pain, pain during limb manipulation, exercise-associated lameness and ground force plate	Group II showed a reduction in overall pain (77%), pain upon limb manipulation (83%) and exercise-associated lameness (84%) (*p* < 0.05) Group II peak vertical force (PVF) was elevated from 7.467 ± 0.419 to 8.818 ± 0.290 N/kg b.w., and the impulse area was elevated from 1.154 ± 0.098 to 1.670 ± 0.278 Ns/kg b.w
Gupta et al. 2012 [58]	Seven to ten client-owned dogs with OA per group - Placebo (Group I) - UC-II^®^ (Group II) - GLU + CHO (Group III) - UC-II^®^ + GLU + CHO (Group IV) 150 days	Evaluate overall pain, pain during limb manipulation, exercise-associated lameness and ground force plate	Group II showed reductions in overall pain (81%), pain upon limb manipulation (87%) and exercise-associated lameness (90%) (*p* < 0.05) Group III exhibited a reduction in overall pain (51%), pain upon limb manipulation (48%) and exercise-associated lameness (43%) (*p* < 0.05) Group IV exhibited a reduction in overall pain (36%), pain upon limb manipulation (34%) and exercise-associated lameness (40%) (*p* < 0.05) Increase in PVF and impulse area in Group II (*p* < 0.05)
Yoshinari et al. 2015 [59]	Twenty client-owned dogs with OA - Placebo (Group I) - NEXT-II (Group II) 150 days	Evaluate overall pain, pain during limb manipulation and pain from physical exertion	Group II showed overall pain reduced by 54.3%, pain upon limb manipulation decreased by 65.2% and pain after physical exertion reduced by 62.5% (*p* < 0.05)
Stabile et al. 2019 [60]	Forty-six client-owned dogs with OA - Robenacoxib (Group I) - UC-II^®^ (Group II) 30 days	Evaluate the clinical scores and mobility based on the Liverpool Osteoarthritis in Dogs (LOAD)	Owner-assessed data showed a similar reduction in the LOAD and mobility scores (*p* < 0.05) of the two groups There were no significant differences between treatments (*p* > 0.05)
Kunsuwannachai and Soontornvipart 2020 [61]	Nine client-owned dogs (13 stifle joints) with OA - Undenatured type II collagen (Group I) 112 days	Evaluate lameness score, radiographic examination scale, ultrasonographic and owner questionnaires: Canine Pain Inventory (CBPI) before and after treatment	Lameness score and radiographic examination showed no difference (*p* > 0.05) Ultrasonographic examination showed differences at 8 and 16 weeks (*p* < 0.05) The CBPI was different (*p* < 0.05)
Varney et al. 2020 [62]	Forty healthy Labrador retrievers - Placebo (Group I) - UC-II^®^ (Group II) 14 days	Evaluated interleukin-6 (IL-6) and cartilage oligomeric matrix protein (COMP) pre and post run	Group II had lower IL-6 and COMP (*p* < 0.05)
Varney et al. 2021 [63]	Forty healthy Labrador retrievers - Placebo (Group I) - UC-II^®^ (Group II) 91 days	Evaluate activity per kilometre and average moving speed Evaluate biomarkers of IL-6, creatine kinase-MM (CKM) and COMP pre- and post-run	Activity per kilometre was greater in Group II vs. group I among males over all runs (*p* < 0.05) Average moving speed was greater in Group II (*p* < 0.05) Group II had significantly lower IL-6 and COMP (*p* < 0.05) No differences found between groups for CKM (*p* > 0.05)
Cabezas et al. 2022 [64]	110 client-owned dogs with OA - UC-II^®^ 10 (Group I) 182 days	Evaluate pain, general condition, appetite, mobility and lameness	Parameters assessed were significantly lower at four, five and six months (*p* < 0.05)
Stabile et al. 2022 [65]	Eighty-six client-owned dogs with OA Four groups: - Placebo (Group I) - Cimicoxib (Group II) - UC-II^®^ (Group III) - Cimicoxib + UC-II^®^ (Group IV) 30 days	Veterinarian evaluation (posture, gait analysis, articular pain, and range of motion), Canine Osteoarthritis Staging Tool (COAST) and owner evaluation LOAD	Reduction in LOAD, mobility scores and clinical scores was recorded in Groups II, III and IV (*p* < 0.05) OAD decreased 29.5% in Group II, 31.4% in Group III and 21.1% in Group IV Reduction in COAST was recorded in Groups II, III and IV (*p* < 0.05)
Varney et al. 2022 [66]	Forty healthy Labrador retrievers - Placebo (Group I) - UC-II^®^ (Group II) 91 days	Evaluated CBPI, LOAD and Gait Analysis Four Rivers Kennel (FRK) inflammatory index score pre-and post-run	Group II had lower points for CBPI and LOAD (*p* < 0.05) Group II had an improved FRK inflammation index score (*p* < 0.05)
Stabile et al. 2022 [67]	Twelve client-owned dogs with OA and 10 without AO - AO free (Group I) - UC-II^®^ (Group II) 30 days	Evaluation of clinical score, mobility score and owner LOAD. Compare the metabolomic synovial fluid	The clinical score, mobility score and LOAD were lower in Group II at 30 days (*p* < 0.05) The values of β-hydroxyisobutyrate, glutamine, creatine and trimethylamine-N-oxide were decreased in Group II at 30 days (*p* < 0.05) Citrate increased in Group II at 30 days (*p* < 0.05)

Abbreviations: Glucosamine (GLU), Chondroitin (CHO), (-)-hydroxycitric acid (HCA) and chromemate (CM).

**Table 6 animals-13-00870-t006:** Literature overview of the evidence regarding *Boswellia serrata* for the management of osteoarthritis (OA) and joint support in dogs.

Reference	Nº Animals, Groups and Duration	Objective	Main Results
Reichling et al. 2004 [34]	Twenty-four dogs with OA or spinal degenerative conditions - *Boswellia serrata* (Group I) 42 days	Evaluate overall efficacy (very good, good, moderate and insufficient) and clinical signs	Overall efficacy was very good and good in 71% at two weeks, 67% at four weeks and 71% at six weeks (*p* < 0.05) Reduction in the severity and resolution of typical clinical signs such as intermittent lameness, local pain and stiff gait were reported at six weeks (*p* < 0.05)
Moreau et al. 2014 [68]	Thirty-two owned dogs with OA Two groups: - Placebo (Group I) - *Boswellia serrata* (Group II-A) - *Boswellia serrata* Group II-B 61 days	Evaluate peak vertical force (PVF) and case-specific outcome measure of disability (CSOM); locomotor activity was recorded using an accelerometer	Groups II-A and II-B had higher a PVF at week four and week eight (*p* < 0.05) No change was observed in CSOM Groups II-A and II-B had higher locomotor activity at week eight (*p* < 0.05)
Manfredi et al. 2018 [52]	Forty-two dogs with hip or elbow dysplasia - Placebo (Group I) - *Boswellia serrata* + GLU + CHO (Group II) +/− 300 days	Evaluate orthopaedic variables (lameness, range of motion, swelling, pain)	Group II had a less severe degree of osteoarthritis at 12 months
Musco et al. 2019 [26]	Twenty dogs with OA - Placebo (Group I) - *Boswellia serrata* + GLU + CHO (Group II) 90 days	Evaluate scores for lameness, pain on manipulation and palpation, range of motion and joint swelling Evaluate metabolic effects	All clinical signs (lameness, pain on manipulation and palpation, range of motion and joint swelling) improved in Group II (*p* < 0.05) Decreased reactive oxygen metabolites and increased biological antioxidants Decreased IL-6 and increased IL-10 (*p* < 0.05)
Muller et al. 2019 [69]	Twenty-two owned dogs with OA - Placebo (Group I) - *Boswellia serrata* + Hyaluronic acid (Group II) 84 days	Evaluate owner questionnaires: Canine Pain Inventory (CBPI) and Liverpool Osteoarthritis in Dogs (LOAD) Evaluate inflammatory biomarkers	CBPI were lower in Group II on days 0 and 84 (*p* < 0.05) LOAD scores were lower in Group II on day 84 (*p* < 0.05) Inflammatory biomarker IL-2 decreased in Group II (*p* < 0.05)
Martello et al. 2019 [54]	Eight owned dogs with OA - *Boswellia serrata* + Cannabidiol (Group I) 30 days	Evaluate efficacy using veterinary and owner questionnaires: Helsinki Chronic Pain Index (HCPI)	The veterinarian and owner agreed on the high effectivity HCPI scores were lower at 30 days (*p* < 0.05)
Caterino et al. 2021 [70]	Twenty owned dogs with OA - GLU + CHO (Group I) - *Boswellia serrata* + GLU + CHO (Group II) 90 days	Evaluate orthopaedic and neurologic signs and force plate gait	Improvement in PVF in 80% of patients, despite no statistical significance between groups
Martello et al. 2021 [71]	Twenty-seven owned dogs with OA - Placebo (Group I) - *Boswellia serrata* (Group II) 60 days	Evaluate C-reactive protein (CRP) and Glutathione (GSH) Owner questionnaire: HCPI	GSH was higher in Group II (*p* < 0.05) CRP was lower in Group II (*p* < 0.05) No change was observed in HCPI (*p* > 0.05)
Gabriele et al. 2022 [72]	Twenty-seven owned dogs with OA - Placebo (Group I) - *Boswellia serrata* (Group II) 170 days	Evaluate CRP and GSH Owner questionnaire: HCPI	HCPI decreased in Group II (*p* < 0.05) GSH was higher in Group II (*p* < 0.05) CRP was lower in Group II (*p* < 0.05)
Cardeccia et al. 2022 [73]	Twenty-four owned dogs with OA - Placebo (Group I) - Boswellia gum (Group II) 60 days	Evaluate CBPI and Hudson activity scale (HAS)	No differences were observed in CBPI or HAS between groups (*p* > 0.05)

Abbreviations: Glucosamine (GLU), Chondroitin (CHO).

**Table 7 animals-13-00870-t007:** Literature overview of evidence regarding the combination of Undenatured Type II Collagen and *Boswellia serrata* for the management of osteoarthritis (OA) and joint support in dogs.

Reference	Study Design and Duration	Objective	Main Results
Martello et al. 2018 [36]	Thirteen client-owned dogs with OA - Undenatured Type II Collagen + *Boswellia serrata* + GLU + CHO (Group I) 60 days	Conduct a general examination and orthopaedic examination and evaluate degree of lameness, overall classification (failure, improved, success) and Helsinki Chronic Pain Index (HCPI)	In 84% of patients, the clinical status was “improved”, in 8% it was “successful” and in 8% it “failed”. In 69% of patients, the degree of lameness improved Dogs with an HCPI between 11 and 19 had a final score similar to their initial score The three cases with the highest index (HCPI > 20) showed a final score that was reduced significantly after 60 days of treatment
Martello et al. 2022 [37]	Forty client-owned dogs with OA - Placebo (Group I) - Undenatured Type II Collagen + *Boswellia serrata* + GLU + CHO (Group II) 60 days	Evaluation of clinical signs of OA by the veterinarian and chronic pain questionnaires using HCPI by the owner	An improvement in signs of OA was recorded in Group II (*p* < 0.05) Lower scores for HCPI were found in Group II (*p* < 0.05)

Abbreviations: Glucosamine (GLU), Chondroitin (CHO).

## 4. Discussion

This article is the first scoping review that compiles the results obtained in studies that used undenatured type II collagen, *Boswellia serrata* or both as supplementary feeds for managing OA and in dogs. After reviewing the currently available literature, it was observed that there are a limited number of studies focused on this topic, although OA has been managed for almost two decades with both undenatured type II collagen and *Boswellia serrata*. However, their combined use for managing OA is novel and has become very popular in recent years, with a total of two articles appearing in the last four years [36,37]. All the studies obtained for the review are prospective, and more than two-thirds of them have a high level of evidence (LoE) [74]. From an ethical point of view, it is not reasonable to leave a patient who has been diagnosed with OA without adequate pain management because they belong to the placebo control group. For this reason, some studies either did not have a control group, or the control group were treated with other food supplements or drugs. As for the Discussion, it is organised into three parts, each with a subheading, referring to the tables previously described in the Results section.

### 4.1. Undenatured Type II Collagen

After reviewing the articles, we observed that not all the studies used the same undenatured type II collagen product in their clinical trials. In most studies, it was used under a registered trademark that refers to a patented undenatured type II collagen known as UC-II^®^ (InterHealth Nutraceuticals Inc., Benicia, CA, USA, Flexadin Advanced^®^, Vetoquinol SA, Tarare, France and Lonza Consumer Health Inc., Morristown, NJ, USA) [30,55,56,57,58,60,62,63,64,65,66,67]. The products on the market that contain this undenatured type II collagen include a daily dose of 40 mg that provides 10 mg of active ingredient UC-II^®^. Other studies used undenatured type II collagen referred to as native type II collagen (Confis Ultra, Candioli Pharma S.r.l.) [36,37] or water-soluble Undenatured Type II Collagen (NEXT-II) (Ryusendo Co. Ltd., Tokyo, Japan) [59]. One article does not specify the undenatured type II collagen product that the authors used [61]. In a recent study, the authors found significant differences when comparing the analytical and physicochemical characteristics of feed supplements currently available on the market that contain undenatured type II collagen [75]. However, these differences could be due to the sample analysis methods used, since there is some discrepancy between the authors’ findings [76,77]. Regardless, the authors recommended that this fact be taken into account when interpreting the results in a review, as it may not be possible to extrapolate the benefits obtained with a specific product in a given study to all products labelled with undenatured type II collagen [75].

Of the fourteen published studies that evaluated the effects of undenatured type II collagen administration alone, we found that there are mainly two research groups who have evaluated this feed supplement for the management of OA: five studies were conducted at Murray State University [30,55,56,57,58] and three were conducted at the University of Bari [60,65,67]. This could explain why the methodologies and results obtained by the first research group are so similar to each other. In the study published by Deparle and colleagues [55], the authors obtained a more beneficial management of OA with the use of active ingredient UC-II^®^ at 10 mg/day rather than 1 mg/day, which justifies the use of this dose in subsequent research and commercial products. However, the results of this study should be interpreted with caution, because the owners were blinded to the product assigned to these groups but not the placebo group. Studies comparing the use of UC-II^®^ with other feed supplements, such as GLU, CHO, (-)-hydroxycitric acid (HCA) and chromemate (CM), showed that the former’s use for multimodal pain management may be one of the most suitable options [30,56,58]. The use of UC-II^®^ for managing OA in dogs can provide sufficient clinical benefits on its own, without combinations with other complementary feeds being necessary, as there is no apparent improvement in the pain scales or ground force plate scores [58]. UC-II^®^ was compared with the use of the drug par excellence for the management of OA, NSAIDs, by Stabile and colleagues in two studies. Although equipotency with NSAIDs was not demonstrated, when comparing by subgroup and using the Liverpool Osteoarthritis in Dogs (LOAD) scale, the owners observed a reduction in pain similar to that obtained by Robenacoxib or Cimicoxib in the initial stages of OA [60,65]. However, the results of the Robenacoxib study should be viewed with caution due to the lack of a control group and the fact that the owners were aware of their pets’ treatment, which may have caused a placebo effect among the owners [60].

In patients with OA that is secondary to patellar luxation, the use of undenatured type II collagen does not produce an evident reduction in pain-associated lameness, but an improvement in quality of life is observed based on the Canine Brief Pain Inventory scale; hence, it should be considered for cases where surgical intervention is not an option [61]. These results differ from those obtained by the Murray State University research group [30,55,56,57], which may be due to the use of different scales to assess pain-associated lameness or the fact that the margin of improvement is narrower in patients with OA associated with patellar luxation. Other techniques have been used to evaluate the effects of undenatured type II collagen on patients with OA, such as radiographic examination, with no differences observed pre- and post-supplementation, because in its early stages, the diagnosis of OA by radiography is complicated. In contrast, ultrasound examination is a tool for evaluating synovial fluid, which has a better appearance after undenatured type II collagen treatment [61] and may suggest an improvement in inflammation as a result of joint effusion. These results agree with those published by Stabile and colleagues [67], who evaluated inflammatory biomarkers in the synovial fluid. A recurrence of signs of OA was documented after the cessation of supplementation with UC-II^®^ [55,56], which reaffirms the benefits of managing patients with this feed supplement. Long-term supplementation with UC-II^®^ showed no adverse effects on, or significant changes in, biochemical parameters, suggesting that its administration is well tolerated and safe [57,58,63]. Even in response to increased pain-reducing activity, patients on the UC-II^®^ supplement showed a more adequate body weight [30]. Varney and colleagues investigated the use of UC-II^®^ with the aim of decrease joint inflammation after exercise, and they observed a higher activity per kilometre and less response from inflammatory biomarkers in dogs [62,63,66]. 

### 4.2. Boswellia serrata

*Boswellia serrata*, as a feed supplement, has begun to be used in veterinary medicine for patients with OA due to the good results obtained for humans, in whom it has been observed to act as an anti-inflammatory, anti-arthritic and analgesic agent [78,79]. In the present review, all the articles, with the exception of one [34], evaluated the use of *Boswellia serrata* together with other food supplements to manage OA in dogs. This makes it difficult to evaluate the specific benefits that *Boswellia serrata* can provide in managing OA, since we cannot discern the influences that other feed supplements have on the results obtained in these studies. In addition, it has been observed that some of the articles do not explicitly describe the dose or the percentage of *Boswellia serrata* used for the study; thus, the comparison between the different studies that used *Boswellia serrata* extracts (boswellic acids, resin, AKBA) is controversial due to variations in the purity and compositions of the products used. What the studies all seem to have in common, however, is the combined use of the product with other feed supplements or natural herbs, showing a reduction in chronic pain, improving clinical signs, especially in severe cases, and reducing inflammatory biomarkers [26,34,36,68,69,70,71,72]. The decreases in alkaline phosphatase, cholesterol and triglycerides [26] can be explained by taking into account the fact that triglycerides and cholesterol can be increased due to the inflammatory process that results in OA and reduced mobility [80], thus demonstrating the beneficial effect of using these feed supplements for pain management and increased physical activity.

### 4.3. Undenatured Type II Collagen and Boswellia serrata

The authors found two articles describing studies in which undenatured type II collagen and *Boswellia serrata* were used together with other feed supplements (GLU, CHO and Hyaluronic acid) but did not find any studies that used the two alone. One study was a randomised controlled trial [37], while the other one did not have a control group was not randomised or blinded [36], and the bias that this may imply for the interpretation of the results must be considered. These studies focused on the management of OA and, to date, there is nothing published on the treatments’ combined use for to decrease inflammation or degeneration of the cartilage joints. The fact that they have been combined with other food supplements implies that a conclusion cannot be drawn about the effects obtained using undenatured type II collagen and *Boswellia serrata* alone, since the authors did not know how GLU, CHO and Hyaluronic acid influenced the results. However, the results obtained are similar to those observed after the use of UC-II^®^ for patients with OA in cases where CHO, GLU and Hyaluronic acid were not used. The use of UC-II^®^ seems to be more effective for more severe degrees of OA, which may be due to a greater margin for improvement [37]. It is known that it can take up to three months for the oral tolerance mechanism to be fully activated [47]; thus, the maximum efficacy of undenatured type II collagen may not yet be reached at 60 days. This hypothesis is consistent with the results obtained by the Murray State University research group, who obtained higher percentages for the reduction in OA-associated pain when they treated patients for longer periods of time [30,56,57,81]. 

## 5. Conclusions

In conclusion, most of the records published in the veterinary literature on the use of undenatured type II collagen for the management of patients with clinical signs of OA used a study design with a high LoE, which means that the results obtained are reliable. The treatment improves the general condition and appetite, reduces the degree of lameness, and increases physical activity or mobility at the same time that it produces changes on the metabolic level and in the synovial fluid, enabling practitioners to consider it as an option for decrease joint inflammation during intense exercise and upon the appearance of clinical signs of diseases that predispose an individual to OA. On the other hand, the evaluation of the effects produced by *Boswellia serrata* is more complicated, because it is always combined with other feed supplements, and the compositions of the products are different, but in general terms, it produces benefits in relieving pain and reducing the clinical signs of OA in dogs. Finally, the combination of both with other feed supplements in the same product provides results similar to those obtained in studies of UC-II^®^. The combination of undenatured type II collagen and *Boswellia serrata* alone has not been investigated for the management of OA and helps support joint health in dogs. Its tolerance and low adverse effects render it a feed supplement that can control pain associated with OA in the long term. Due to the slow onset of action of undenatured type II collagen, the use of this feed supplement should not be considered as the first-line option when an immediate analgesic effect is sought. 

## Figures and Tables

**Figure 1 animals-13-00870-f001:**
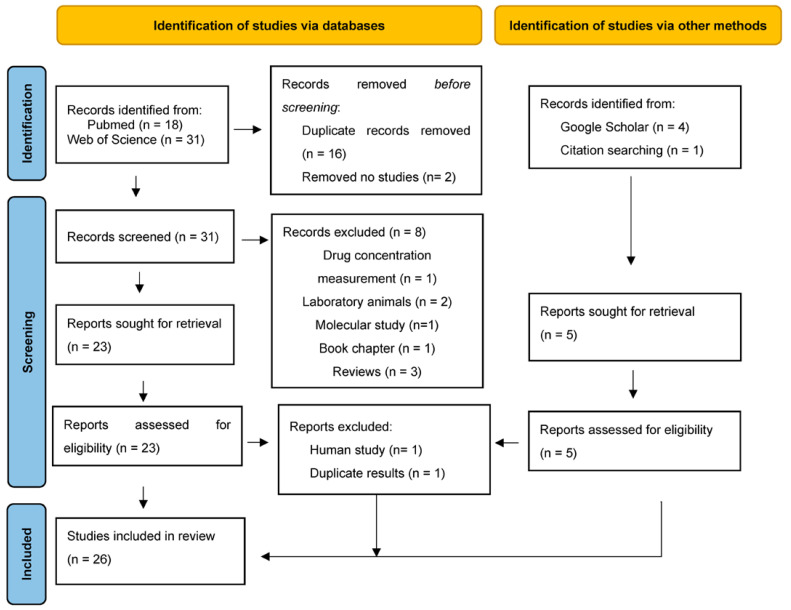
Flowchart of the selection process for scientific records to be analysed in this review.

**Table 1 animals-13-00870-t001:** Records were categorised according to the scientific study design.

Type	Description Study Design
I	Prospective Randomised Placebo-Controlled Clinical Trial
II	Prospective Randomised Controlled Clinical Trial
III	Prospective Clinical Trial

**Table 2 animals-13-00870-t002:** Records included in the review based on the search performed regarding undenatured type II collagen for managing clinical signs of osteoarthritis and joint support in dogs.

	Records of UC-II	Year of Publication	Study Design	Blind	Ref.
1	Efficacy and safety of glycosylated undenatured type-II collagen (UC-II) in therapy of arthritic dogs	2005	Type I	Yes	[55]
2	Therapeutic Efficacy and Safety of Undenatured Type II Collagen Singly or in Combination with Glucosamine and Chondroitin in Arthritic Dogs	2007	Type I	Double-blind	[30]
3	Therapeutic efficacy and safety of undenatured type-II collagen (UC-II) alone or in combination with (-)-hydroxycitric acid and chromemate in arthritic dogs	2007	Type I	No information	[56]
4	Suppression of arthritic pain in dogs by undenatured type-II collagen (UC-II) treatment quantitatively assessed by ground force plate	2009	Abstract congress Type I	No information	[57]
5	Comparative therapeutic efficacy and safety of type-II collagen (UC-II), glucosamine and chondroitin in arthritic dogs: pain evaluation by ground force plate	2012	Type I	Double-blind	[58]
6	An Overview of a Novel, Water-Soluble Undenatured Type II Collagen (NEXT-II)	2015	Type I	Yes	[59]
7	Evaluation of the Effects of Undenatured Type II Collagen (UC-II) as Compared to Robenacoxib on the Mobility Impairment Induced by Osteoarthritis in Dogs	2019	Type I	Yes	[60]
8	Chondroprotective efficacy of undenatured collagen type II on canine osteoarthritis secondary to medial patellar luxation	2020	Type II	No	[61]
9	Undenatured type II collagen mitigates inflammation and cartilage degeneration in healthy untrained Labrador retrievers after exercise	2020	Abstract congress Type I	No information	[62]
10	Undenatured type II collagen mitigates inflammation and cartilage degeneration in healthy Labrador Retrievers during an exercise regimen	2021	Type I	No information	[63]
11	Long-term supplementation with an undenatured type-II collagen (UC-II) formulation in dogs with degenerative joint disease: Exploratory study	2022	Type III	No	[64]
12	Evaluation of clinical efficacy of undenatured type II collagen supplementation compared to cimicoxib and their association in dogs affected by natural occurring osteoarthritis	2022	Type I	Yes	[65]
13	Impact of supplemented undenatured type II collagen on pain and mobility in healthy Labrador Retrievers during an exercise regimen	2022	Type I	No information	[66]
14	^1^H-NMR metabolomic profile of healthy and osteoarthritic canine synovial fluid before and after UC-II supplementation	2022	Type II	No	[67]

**Table 3 animals-13-00870-t003:** Records included in the review based on the search performed regarding *Boswellia serrata* for managing clinical signs of osteoarthritis and joint support in dogs.

	Records of *Boswellia serrata*	Year of Publication	Study Design	Blind	Ref.
1	Dietary support with Boswellia resin in canine inflammatory joint and spinal disease	2004	Prospective multicentre Type III	No	[34]
2	A medicinal herb-based natural health product improves the condition of a canine natural osteoarthritis model: a randomized placebo-controlled trial	2014	Type I	Double-blind	[68]
3	Effect of a commercially available fish-based dog food enriched with nutraceuticals on hip and elbow dysplasia in growing Labrador retrievers	2018	Type I	Yes	[52]
4	Effects of a nutritional supplement in dogs affected by osteoarthritis	2019	Type I	Double-blind	[26]
5	Placebo-controlled pilot study of the effects of an eggshell membrane-based supplement on mobility and serum biomarkers in dogs with osteoarthritis	2019	Type I	Double-blind	[69]
6	Effects on Pain and Mobility of a New Diet Supplement in Dogs with Osteoarthritis: A Pilot Study	2019	Type III	No	[54]
7	Clinical efficacy of Curcuvet and Boswellic acid combined with conventional nutraceutical product: An aid to canine osteoarthritis	2021	Type II	Double-blind	[70]
8	Preliminary results on the efficacy of a dietary supplement combined with physiotherapy in dogs with osteoarthritis on biomarkers of oxidative stress and inflammation	2021	Type I	Double-blind	[71]
9	Long-term effects of a diet supplement containing *Cannabis sativa* oil and *Boswellia serrata* in dogs with osteoarthritis following physiotherapy treatments: a randomised, placebo-controlled and double-blind clinical trial	2022	Type I	Double-blind	[72]
10	A pilot study examining a proprietary herbal blend for the treatment of canine osteoarthritis pain	2022	Type I	Double-blind	[73]

**Table 4 animals-13-00870-t004:** Records included in the review based on the search performed regarding the combination of undenatured type II collagen and *Boswellia serrata* for managing clinical signs of osteoarthritis and joint support in dogs.

	Records of Combined UC-II and *Boswellia serrata*	Year of Publication	Type of Record	Blind	Ref.
1	Evaluation of The Efficacy of a Dietary Supplement in Alleviating Symptoms in Dogs with Osteoarthritis	2018	Type III	No	[36]
2	Efficacy of a dietary supplement in dogs with osteoarthritis: A randomized placebo-controlled, double-blind clinical trial	2022	Type I	Double-blind	[37]

## Data Availability

The data presented in this study are available in the studies included in this systematic review.

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
