# Peer review of "Management of Osteoarthritis and Joint Support Using Feed Supplements: A Scoping Review of Undenatured Type II Collagen and Boswellia serrata"

_animals, 2023, doi:10.3390/ani13050870_

Round 1
Reviewer 1 Report
Multimodal Management of Osteoarthritis Using Feed Supplements: A Review of Undenatured Type II Collagen and Boswellia Serrata
General comments
This Literature Review covers an interesting topic in Veterinary Medicine and in particular the use of alternative medicine to manage Osteoarthritis in dogs.
I am aware the Authors did not perform a Systematic Review and meta-analysis. However, given the systematic approach described, I see limitations in the adopted protocol as I have underlined in the comments on the Methods section.
There is a bit of confusion between Results and Discussion, then to be reviewed.
Need to re-check the use of the correct Table format/theme font/bold throughout the Manuscript.
I would simply use “Literature Review” rather than “literature structure/ structured review/ Systematic Review” through the text.
I would suggest to review the English language throughout the text and to re-check the structure of some sentences.
Title
In the title the authors used ”Multimodal Management”. The multimodal approach to OA is related to the use of different therapies including natural supplements, physiotherapy, hydrotherapy, …. So, I would suggest to review the title.
Simple Summary
It should include some results and the conclusion limiting the general information as not so relevant in this section.
Abstract
-Change “ Multimodal management of OA”in “Multimodal management of Osteoarthritis”
-The reported results and details are not homogenous for the 4 options considered (Undenatured Type II Collagen alone, Boswellia serrata alone, Boswellia serrata combined with other feed supplements ,combination Undenatured Type II Collagen and Boswellia serrata). Please, review the entire abstract in order to present the results with similar level of details -when possible.
-“Undenatured Type II Collagen offers better results for managing pain associated with mild to moderate OA than other feed supplements in dogs.” Have you measured that in order to write this statement? If not, please change the statement
-I would replace this sentence “ from which a total of 18 records were obtained”with “from which a total of 18 records were included in this Review”.
Introduction
-It is well written and comprehensive of the major points.
-Please, change: “in canine species” with “in dogs”
Methods
- The search was conducted in PubMed and Web of Science only. Given the limited number of papers found and the conclusions drawn from them, I would suggest to extend the search, using the same approach used for the other two databases, to CAB Abstracts. This database is relevant in Veterinary Medicine and would probably give new hits to check.
In addition, I would strongly suggest to check the reference list of the studies included in this Review to avoid missing relevant articles as some of them could have been published in non-Indexed journals.
Then, I cannot see included in the Review the following relevant studies on this topic you can easily find in Google Scholar (typing “boswellia AND dog”) as you have also reported in the Methods:
* Martello et al. (2019) Effects on Pain and Mobility of a New Diet Supplement in Dogs with Osteoarthritis: A Pilot Study. Ann Clin Lab Res Vol 7. No. 2: 304.
* Martello et al. (2021) Preliminary results on the efficacy of a dietary supplement combined with physiotherapy in dogs with osteoarthritis on biomarkers of oxidative stress and inflammation, Italian Journal of Animal Science, 20:1, 2131-2133.
- This is not clear. Please clarify why.
Pubmed was searched across all fields for Undenatured Type II Collagen and Boswellia serrata, while title/abstract was used for osteoarthritis and dog
- This is not clear. Please clarify why.
The Web of Science search was performed by topic.
-There is a specific reason why you chose 2004? Please, explain
“a time filter from January 2004 to July 2022”
-This statement is generic. Could you please specify you decided to use Google Scholar and maybe you checked the first 200/300/? hits or similar, please?
“the articles in Google Scholar were consulted, that were not identified by the search strategy described above.”
-It would have been interesting to include this as well even if it is very uncommon to see papers where only adverse events are described.
“Studies evaluating only adverse effects of the administered feed supplement”
-Please, could you clarify why you used this tool while you are not performing a proper Systematic Review?
“Studies were assessed for the level of evidence (LoE) based on a modified grading 175 system of Winona State University [54]”
-Figure1. Change “tittle” in “title”. It is not clear the last box on the right. Please, replace it with paper included by full text. In general, even if it is not a Systematic Review, I would suggest the use of the PRISMA flow chart as this Figure is quite similar to that.
-Inclusion criteria. Please, explain better what you mean by:
(2) Evaluate their effects in managing or preventing OA.
Which are the parameters you would like the included paper to report in order to be included? Chronic Pain level? Or?
-Please, list the data you are collecting from each included paper in your data extraction file.
Results
Please, check this sentence:
“the tittle and abstract were reviewed, eliminating articles that did not meet the inclusion and exclusion criteria (n = 8)”
-Line 190-195. I would delete these lines as I would also delete this information in Figure 1.
-Tables are informative but too big and not readable. Please check the tables and use the same structure/ font/ etc. I would suggest to use First Author name and the Reference in parenthesis rather than the full title. You do not need the column “control group” and “randomized” if you do have the column study design, “the blind”column is not clear why the authors use yes / double blinded/triple blinded-please specify the meaning.
-When possible, please, report the statistical analysis performed in the included paper- Maybe in one of the tables.
Discussion
-Tables 5 to 7 should better summarize the information even if all the relevant data are included at this point by the authors. The tables in this format are not easily readable in the main text. The authors should try to move some information in a supplementary file table and the main results of each paper in a table in the Results section in the main text. The authors should also review the discussion as to provide comments on these results without repeating the results themselves.
I think there is no need to report the dose of the supplement in the main table, while you can leave the information in the Supplementary.
The column “study design” does not report the study design in this format.
-Please, double check if the information provided in the Discussion is not part of the Results. Please, try to limit reporting results in this section and focus the attention to discussing the overall results of the included papers. It would be fine if the Discussion is a bit shorter than the one you have now but it must include the most relevant discussion points.
-Please, make more clear in the text the results related to “managing and preventing” Osteoarthritis (OA) in dogs as the difference is not well specified in the text. I can only see data from ref [63 ] which is referred to “OA prevention”.
-Line 375 : change “the other one do not has a control group [37]”to the other one does not have a control group [37]”
-Please,
Replace “In the case of Martello [37,38], he used…”with : “In the papers published by Martello and colleagues [37,38], they have used ….”. Check in the whole manuscript the use of “First author name and colleagues”.
-I strongly recommend the authors to limit the information on Flexadin® Advanced (Vetoquinol S.r.l) as the reference to unpublished data and to a product on the market that has not been tested in a scientific trial sounds inappropriate in this context.
-Please, use Candioli Pharma Srl through the text
-Please, use Confis Ultra (Candioli Pharma Srl) and not b-2Cool® , Confis ultra
Conclusion
Please, write the conclusion of the Review
Data Availability Statement: Please, change Systematic Review into Literature Review
“The data presented in this study are available in the included studies of this systematic review.”
Reviewer 2 Report
Please clarify these questions:
- L26-27: Author´s wrote In conclusion, Undenatured Type II Collagen offers better results for managing pain associated with mild to moderate OA than other feed supplements in dogs. But, this conclusion does not match with the conclusion section at the end of the article (L417-421).
Also, related the conclusión described in L26-27. Please, justify this conclusion described in the Abstract section. Do the authors believe that there is scientific evidence for this conclusion?. Do the authors believe that they have shown enough scientific evidence in this review paper for this conclusion?
This reviewer considers the conclusion described between L417-420 more acceptable.
- L35: This reviewer considers it inappropriate to include the term UC-II® within the keywords because, UC-II® is a registered trademark of a type of undenatured type II collagen, and also the inclusion of a registered trademark as a Keyword is not suitable.
- Table 4: The article described in the second is named with the number 3, it should be a 2.
- L150: The term UCII is cited for the first time without been previously explained.
- L166-L168: Why studies evaluating only alterations in inflammatory biomarkers and/or synovial fluid were excluded in this revision article?. Using this exclusion criteria, this review could miss important information derived from studies that evaluated objective parameters in osteoarthritis.
Round 2
Reviewer 1 Report
I have to acknowledge the great work performed by the Authors reviewing their paper and how well they have addressed my comments.
Just a few more things to note.
Please, make sure you use italics for Boswellia serrata.
Please, review this sentence in the Summary: “Through this scoping review are exposed 26 studies published”
Please, being the manuscript now defined as a Scoping review, it would be great if you specify in the methods that you have double screened title and abstract for the 10% of the papers and that your level of agreement was xx%, then a single reviewer performed the rest of the screening.
